# Improving Model Merging with Natural Niches

**João P. Abrantes, Robert Tjarko Lange, Yujin Tang**
Sakana AI
{joao,robert,yujintang}@sakana.ai

## Abstract

Model merging is a powerful technique to combine specialized knowledge of multiple machine learning models into a single unified model. However, current methods require manually partitioning the model parameters into a fixed number of groups to be merged, which constraints the exploration of potential combinations and limits performance. To address these limitations, we propose an evolutionary algorithm with three key features: (1) dynamically adjustment of merging boundaries to progressively explore a broader range of parameter combinations; (2) a diversity preservation mechanism inspired by nature, which maintains a population of diverse, high-performing models that are particularly effective for merging; and (3) a heuristic-based *mate selection* strategy to identify the most promising pairs of models for merging. Our experimental results show, for the first time, that model merging can be used to evolve models from *scratch*. Specifically, we evolve MNIST classifiers from scratch using our method, and achieve comparable performance to CMA-ES, while being computationally cheaper. Additionally, we use our method to merge specialised language models and obtain state-of-the-art performance. Our code is available at `https://github.com/AnonScientist/natural_niches`.

## 1 Introduction

Open-source generative models have allowed the proliferation of thousands of specialised variants, fine-tuned by practitioners to solve their specific needs. In such an environment, where numerous diverse models are freely accessible, the ability to merge and aggregate that wealth of knowledge into a single model becomes important. This process, known as model merging [14], has gained popularity as evidenced by the current prevalence of merged models on the Open LLM Leaderboard [12].

Model merging initially relied on manually adjusting coefficients to combine seed models, a process guided by intuition and requiring significant trial and error to optimize performance for specific tasks. Recently, this has been streamlined with evolutionary algorithms that automatically search for optimal coefficients [1] and increase the merging efficiency.

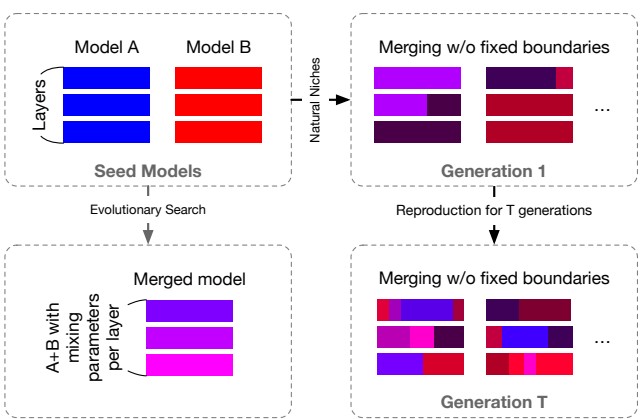

Figure 1: Left, previous methods group the parameters of each seed model according to fixed boundaries (e.g., model layers) and then search for a set of coefficients to mix each group. The shades of purple in the layers of the merged model represent how much the interpolation is close to parent A (blue) or parent B (red). Right, evolution of an archive of models using a random split-point explores a progressively larger number of coefficients and boundaries.

Preprint. Under review.

However, one manual step persists: developers must group model parameters into fixed sets before merging, which restricts the search for potential combinations (see Figure 1 left). To overcome this, we propose an evolutionary algorithm with three key features:

**1. Evolving the Merging Boundaries**. Existing methods partition the parameters of each seed model into fixed groups (e.g., layers) and search for optimal coefficients for merging these groups, which confines exploration to predefined boundaries. In contrast, our approach merges two models at a time, using arbitrary split points to divide parameters. Rather than working with fixed models, we maintain an evolving archive of models. As the number of generations increases, the method progressively explores a broader set of boundaries and coefficients (see Figure 1 right), allowing for increasingly complex combination if needed. This optional and gradual increase in complexity ensures a wider range of possibilities while maintaining computational tractability.

**2. Managing Diversity**. Merging models only makes sense when they differ, making it essential to maintain diversity within the evolving population. However, the challenge of diversity preservation lies in determining which characteristics should be diverse. While many approaches require developers to manually define a diversity metric, we employ a nature-inspired method to automatically preserve diverse high-performing models that are particularly effective for merging.

**3. Matchmaker**. We introduce a heuristic for pairing models based on their complementary strengths, enhancing both the efficiency and quality of our method. *Mate selection* remains an under-explored area in genetic algorithms, becoming increasingly important as computational costs of crossovers (merging) rise. This work encourages further research on this topic.

We name our method *Natural Niches* and show that it performs well across two vastly different experiments: 1) evolving small classifiers *from scratch* and from pre-trained models (section 4.1), and 2) merging Large Language Models (LLMs) (section 4.2).

## 2 Related Work

### 2.1 Model Merging

Model merging introduces an innovative approach for integrating the strengths of multiple pre-trained models. In contrast to fine-tuning, which focuses on refining a single pre-trained model, model merging can leverage several models concurrently without requiring backpropagation. This has allowed the method to combine extremely large models for tasks involving subjective goals, like customizing an image generation model to reflect personal tastes.

Notably, the release of Stable Diffusion (SD) [21] and open-source interfaces [3] enabled practitioners to merge different SD fine-tunes manually, using techniques like linear and spherical linear interpolation (SLERP) [25]. These early efforts demonstrated the potential of model merging in combining specialized capabilities into a single unified model.

Subsequent research has approached the model merging problem from two complementary directions: minimizing interference between models and automating the merging process. Methods such as TIES [29] and DARE [31] introduced strategies to balance the contributions of individual models while minimizing interference, ensuring that the strengths of each model are preserved without mutual disruption.

Evolutionary algorithms like CMA-ES [11] were later applied to automate the search for optimal merging coefficients. As explored in [1], these methods not only automate what was previously a manual, iterative process but also significantly improve merging efficiency.

While previous research centered on merging pre-trained models, we show that merging can efficiently be used to evolve models from scratch. Additionally, unlike earlier methods that required manual partitioning of model parameters, we automate and optimize this process during the evolutionary process.

### 2.2 Overview of Diversity Preservation in Genetic Algorithms

Diversity preservation in Genetic Algorithms (GA) is crucial for finding multiple solutions to multimodal problems [26] and to prevent premature convergence. We believe this is particularly important when using crossover operations (such as model merging). These operations benefit

from diversity while at the same time reducing it, which may lead to premature convergence if not counter-acted by a diversity increasing mechanism. In this section, we provide a quick overview of the two main methods for diversity preservation in GA: 1) crowding [7, 27] and 2) fitness sharing [10, 8, 9, 20].

**Crowding methods** involve first applying mutation and crossover to produce new candidate solutions. These candidates then compete for inclusion in the population, but only against other candidates that are similar, based on a predefined criteria such as genetic or phenotypic distances. This selective competition helps maintain diversity within the population by preventing any single solution type from becoming overly dominant. A similar mechanism for selective competition is used in the popular algorithm of MAP-Elites [18]. In MAP-Elites, the solution space is divided into a multidimensional grid, with each cell representing a species defined by one or more predefined behaviour descriptors. New candidates are placed into cells based on their descriptors and replace existing solutions only if they perform better. The real challenge of this method lies in defining descriptors that promote the desired type of diversity.

**Fitness sharing** requires each individual to share its rewards with others. In *explicit* fitness sharing, the researcher defines a distance function that is used to cluster similar individuals into a species, each individual then shares its fitness with other members of its species, making it more difficult for any single species to grow excessively large. A notable example is the NEAT [24] algorithm, known for evolving neural networks topologies, which clusters solutions into species by measuring genotypic differences (distances in network topologies). *Implicit* fitness sharing [23, 6], is seen as the more natural method because, as in Nature, it protects niches rather than species. A niche is a group of individuals that compete for the same resources, while a species is defined as group that can interbreed and typically have small genetic and phenotypic differences. Usually, members of the same species compete for the same resources (e.g., food, partners, shelter), but vastly different species can also compete for the same vital resources like nesting sites or food sources (e.g., birds and bats, lions and hyenas). *Implicit* sharing does not rely on custom distance metrics. Instead, it simulates natural competition for limited resources, promoting diversity as individuals who can derive their fitness from less contested resources are favoured. We provide more details on Section 3.

## 3    Natural Niches

In model merging, the goal is to find the optimal parameters $\theta^*$ for a merged model from a set of $K$ seed models, each of which characterized by its model parameters $\theta_i$ $(i = 1 \cdots K)$, such that optimization goal, normally in the form of task scores summation or average, is maximized. The following equation expresses this description mathematically:

$$\theta^* = \arg\max_\theta \sum_{j=1}^{N} s(x_j \mid \theta), \text{where}, \theta = h_w(\theta_1, \cdots, \theta_K) \tag{1}$$

where $h_w$ is the model merging function parameterized by $w$'s that correspond to fixed model merging boundaries (e.g., one scalar $w_{k,l}$ for the $l$-th layer in the $k$-th seed model), $s$ is the score function for a certain task, $x_j$ is a task example, and $N$ is the number of examples to be evaluated. In Natural Niches (NN), we propose modifications to the merging function $h$ to enable evolution of the merging boundaries, and adjustments to the optimization goal to promote diverse solutions.

### 3.1    Eliminating Fixed Model Merging Boundaries

In the formulation above, finding $\theta^*$ boils down to searching for the optimal model merging parameters $w$ in $h_w$. To get rid of the constraints of fixed model merging boundaries and thus allows more flexibility, we propose to include these boundaries together with the mixing parameters into the evolutionary process. Concretely, NN maintains an archive of models, which is initialized with the $K$ seed models. At each training step, NN randomly picks two models $A$ and $B$ from the archive, and samples two parameters $(w_m, w_s)$ that determines the mixing ratio and the split-point in the models' parameters space. It then merges models $A$ and $B$ with the following formula, and inserts the new model into the archive if it outperforms the worst individual.

$$h_{\text{NN}}(\theta_A, \theta_B, w_m, w_s) = \text{concat}\big(f_{w_m}(\theta_A^{<w_s}, \theta_B^{<w_s}), f_{1-w_m}(\theta_A^{\geq w_s}, \theta_B^{\geq w_s})\big) \tag{2}$$

Here, $\theta^{<w_s}$ and $\theta^{\geq w_s}$ indicate the sub-arrays of model parameters before and after the split-point indexed by $w_s$. $f_t(\theta_A, \theta_B)$ is a spherical linear interpolation of rotations (SLERP) function that interpolates $(\theta_A, \theta_B)$ with $t$. As shown in the right part of Figure 1, our method incrementally expands the search space by exploring a broader set of boundaries and coefficients. This gradual introduction of complexity ensures a wider range of possibilities while maintaining computational tractability.

### 3.2 Encouraging Diversity via a Modified Optimization Goal

Competing for limited resources **naturally** promotes diversity, favoring individuals who can tap into less contested resources. In the context of the optimization goal in Equation 1, where a sum of scores from all the examples is being maximised, each score is a "resource" that contributes to the fitness of a solution. By limiting the resource supply, NN sparks competition which naturally favors individuals that take over new niches. Concretely, we limit the total fitness a population can extract from a data point $x_j$ by a capacity $c_j$. The amount of fitness a candidate solution derives from a data point is proportional to its score relative to the aggregate score of the population. Our modified goal becomes:

$$\theta^* = \arg\max_\theta \sum_{j=1}^{N} \frac{s(x_j \mid \theta)}{z_j + \epsilon} c_j, \text{where, } z_j = \sum_{k=1}^{P} s(x_j \mid \theta_k) \tag{3}$$

where $\epsilon$ in the denominator is a small number to prevent the zero-division error. In the term that defines $z_j$, $P$ is the archive size. The capacity $c_j$, is task dependent and can be defined in multiple ways. For example, in binary scoring tasks (i.e. $s(\cdot) \in \{0, 1\}$) we simply set $c_j = 1$. In one experiment (WebShop) the environment offers a continuous reward from 0 to 1. Here, we define $c_j = \max_i s(x_j|\theta_i)$ to ensure that partially solved data points (where $\max_i s(x_j|\theta_i) < 1$) do not distribute the same amount of fitness points as fully solved data points (where $\max_i s(x_j|\theta_i) = 1$).

### 3.3 Sampling Parents via Matchmaker

Many evolutionary algorithms use the crossover operation to combine the strengths of both parents. In biology, this combination (i.e., reproduction) is very expensive, and therefore, animals invest many resources in the process of mate selection. We believe that as we make use of more expensive crossover operations, like model merging, algorithms for mate selection become increasingly important.

In contrast to conventional methods that put more sampling probability mass on top performing models in the archive, NN adds an extra layer of consideration that takes into account the complementarity of the parent models. Specifically, we sample the first parent based on their weighted sum of scores defined in Equation 3, and then sample the second parent based on a "matching score" generated by function $g$ that is specifically tailored for the first parent. The equation below gives the definition of this matching score, it straightforwardly expresses a desire to choose a model B that performs well in the data points where model A performs less well, while giving an extra preference to resources with high capacity $c_j$ and low competition $z_j$.

$$g(\theta_A, \theta_B) = \sum_{j=1}^{N} \frac{c_j}{z_j + \epsilon} \max\left(s(x_j \mid \theta_B) - s(x_j \mid \theta_A), 0\right) \tag{4}$$

## 4 Experiments

We verify the effectiveness of our proposed method on two challenging tasks: First, we evolve image classifiers from scratch and from pre-trained models on the MNIST dataset, and then we scale up the experiment to merging LLMs.

### 4.1 Experiment 1: Evolving MNIST classifiers

#### 4.1.1 Setup

**Model.** The model being optimized is a two-layer feedforward neural network with 19,210 parameters in total. When starting from scratch, we randomly initialize the models. For pre-trained models, we

develop two specialized models: one is trained on digits 0 through 4, and the other is trained on digits 5 through 9.

**Dataset.** We've used the MNIST [15] dataset from `scikit-learn` [19] where each digit is a 8x8 gray scaled image. 80% of the data was used as the training split, and 20% as the testing split.

**Baselines.** For the MAP-Elites algorithm, we used two diversity dimensions to create a 10 by 10 grid: the accuracy of the model in odd and even numbers. When starting from scratch, we use CMA-ES [11] as a baseline, even though it does not perform model merging here. Since the models are randomly initialized, optimizing mixing coefficients alone would be insufficient. Instead, CMA-ES directly optimizes model weights, which incurs a cubic computational cost $O(n^3)$ with respect to the number of parameters. While this method doesn't scale to larger models, it serves as a benchmark for how a popular evolutionary algorithm performs in this experiment. When working with pre-trained models, we use a brute-force search baseline that merges the two seed models by adjusting a mixing coefficient that ranges from 0 to 1 in increments of 0.00001. This baseline is first evaluated on the training data, and the best coefficient is subsequently evaluated on the test data.

**Evolutionary Operators and Variables.** All model merging methods (which excludes CMA-ES) sample a new candidate at a time and decide sequentially whether to insert the candidate into the archive. NN and GA use an archive of 20, sampling each candidate sequentially and deciding whether to insert it, similar to MAP-Elites. MAP-Elites, uses a 10x10 grid, resulting in an archive size of 100. CMA-ES uses a population of 20, sampling and updating its parameters in batches. When starting from scratch all model merging methods use the same mutation operation (Gaussian noise) and the same crossover operation (SLERP with split-point, as described in section 3.1). However, when dealing with pre-trained models, mutation is omitted because random alterations don't scale well to larger models. By avoiding mutation, we can assess which method is most promising for achieving efficient merging in larger pre-trained models.

**Compute Resources.** The 10 independent runs took about 15 hours for CMA-ES, and about 1h for each of the other methods. We ran this experiment using only CPUs.

### 4.1.2 Results

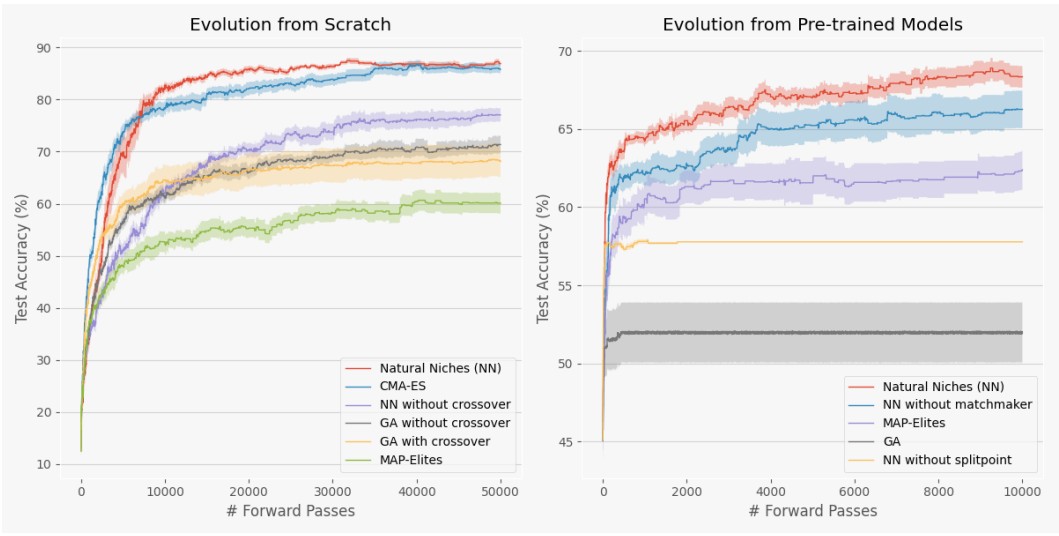

Figure 2: The plots show the accuracy on the test split vs the number of forward passes when starting from randomly initialised models (left) and when starting from the pre-trained models (right). The solid-lines represent the mean of ten independent runs and the shaded area around represents one standard error deviation.

When starting from scratch, NN achieves the highest test accuracy by a substantial margin when compared to the other model merging methods, as shown in Figure 2 (left). Interestingly, GA with crossover performs better early on (before step 12,000) than GA without crossover, however, it converges faster to an inferior solution. The early convergence happens because GA can't maintain a diverse population which is crucial for effective crossover operations. Crossover reduces population

diversity, and without a strong counteracting force, it diminishes exploration. In contrast, NN leverages the crossover operation effectively, benefiting significantly from the diversity it manages to retain. GA is an extreme case where there is no competition, we observed that by progressively decreasing competition in NN, we progressively converge earlier to worse solutions (section 4.1.3). MAP-Elites clusters individuals by their accuracy on odd and even numbers. This means it will always keep individuals who perform poorly on those tasks because there is a slot reserved just for them. Even though those individuals add to the diversity of the population, this is clearly not the type of diversity that leads to strong solutions and it highlights the difficulty of hand-engineering useful diversity metrics.

For models trained from scratch, the split-point and matchmaker have a minimal impact (ablations omitted for clarity). However, as seen in Figure 2 (right), the split-point becomes crucial when starting from pre-trained models, while the matchmaker significantly improves performance throughout the training process. GA has a low average test accuracy with large error bars as its performances is highly dependent on the quality of the first merges. Note that when starting from pre-trained models the mutation operator was not used (as explained in Section 4.1.1), and therefore, the performance is worse.

### 4.1.3 Analysing Diversity and Competition

This section focuses exclusively on the experiment where models were evolved from scratch, as the later sections will provide ample discussion on evolving models starting from pre-trained LLMs.

**Diversity.** Figure 3 left, shows the percentage of training data points that can be correctly labeled by at least one model in the archive, we call this percentage the training coverage. We observe that the archive in NN quickly spreads to cover the majority of the training data points and maintains this high coverage throughout the training process.

The right-hand plot shows how the diversity in the performance of the population evolves with training. If either all models correctly or incorrectly classify a data point, the entropy is 0 (no diversity). In contrast, when the models are evenly split on a prediction, entropy reaches its maximum value of 1. The plot displays the average entropy across all data points. For NN we see a sharp initial rise in entropy followed by a gradual decline as low-performing models go extinct. In contrast, MAP-Elites continually increases diversity by retaining lower-performing models, but it fails to achieve a high coverage. The Genetic Algorithms, lacking a diversity preservation mechanism, reduce coverage early on and show a sharp drop in entropy as they converge prematurely on the best solutions.

Overall, the graphs show that NN maintains an archive of models with complementary strengths that facilitate effective merging, while systematically discarding weaker models as training progresses.

**Competition.** Figure 4 left, shows that smaller archives perform better in the beginning but converge faster to inferior solutions. This suggests that we should scale the archive size along the number of forward passes we want to make. Note that in our plot the computational cost does not increase with the archive size since the number of forward passes remains the same, however, the memory footprint does increase with larger populations. For very large models we can always store the archive on disk instead of keeping them all in the RAM.

For a fixed population size $P$, we can adjust the intensity of competition by introducing a hyper-parameter $\alpha \geq 0$, as described in the fitness function in eq. 5.

$$f(\theta_i) = \sum_{j=1}^{N} \frac{s(x_j|\theta_i)}{z_j^{\alpha} + \epsilon} c_j \tag{5}$$

When $\alpha = 0$, there is no competition because the total fitness available per data point becomes unlimited. When $\alpha = 1$, the total fitness distributed among different individuals is limited to the capacity $c_j$. For $\alpha > 1$, the total fitness distributed decreases with increasing competition ($z_j$), this scenario can be thought of as individuals needing to "fight" for resources, spending some fitness points in the process. Figure 4 right, shows that smaller values of $\alpha$ (i.e. lower competition) have a similar effect to decreasing the population size: it performs better in the beginning but it converges faster to inferior solutions.

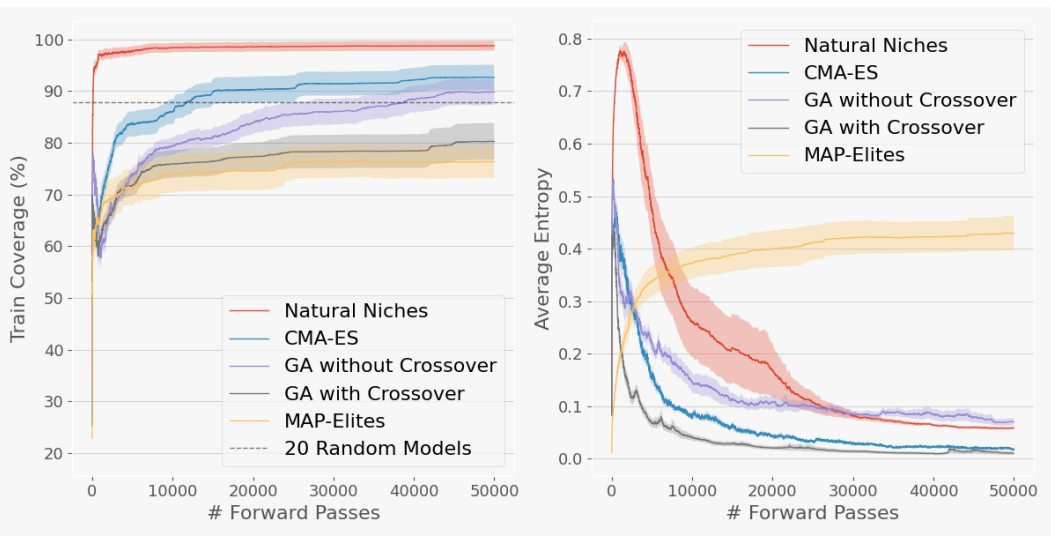

Figure 3: Left: The percentage of training data points that can be correctly labeled by at least one model in the population. Since there are 10 possible labels, 20 random models obtain an average coverage of $1 - (\frac{9}{10})^{20} = 87.8\%$. Right: The evolution of diversity in the population's performance, measured by entropy, over the course of training.

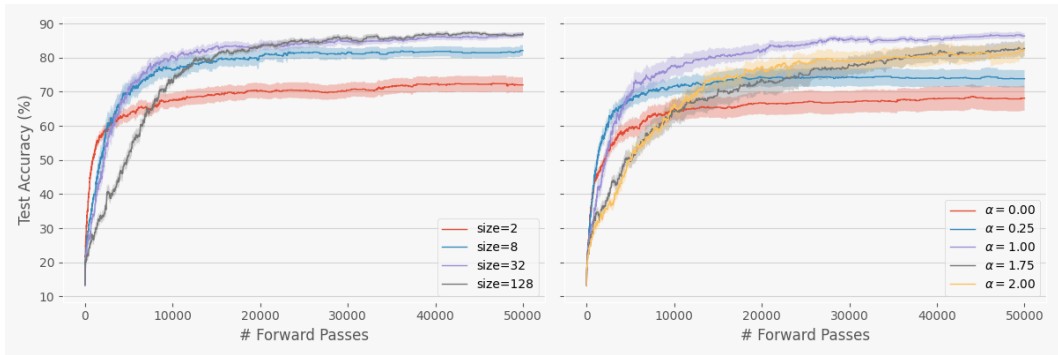

Figure 4: Left, test accuracy of Natural Niches on the MNIST across different archive sizes. Right, test accuracy for different $\alpha$ values while population size is 20.

## 4.2 Experiment 2: LLM Merging

In our LLM Merging experiments we don't use a mutation operator since random mutations don't work well on large models. Moreover, in these experiments we initialise the NN and GA archives with seed models, followed by a short warm-up period (50 iterations or less) where the seed models merge randomly amongst themselves and populate the archive. For these experiments, we used 4 H100 GPUs to run each method for around 24h.

### 4.2.1 Setup: Japanese Math LLM

**Models.** We reproduce the experiment done in [1], where one Japanese specialist LLM, `shisa-gamma-7b-v1` [2], and two Math specialists LLMs, `WizardMath-7B-V1.1` [16] and `Abel-7B-002` [4] are combined to create a hybrid model that can answer math problems in Japanese.

We used exactly the same datasets and evaluation method from [1], but for completeness, we provide the details here.

**Dataset.** The test split, consists of the Japanese test set of the MGSM dataset [22], which is a Japanese translation of a subset of the test set of the GSM8k dataset [5] consisting of 250 samples.

The training set, consists of a translation of the remaining 1069 (out of 1319 examples) of the GSM8k test set that were not included in the MGSM Japanese test set.

**Baselines.** The CMA-ES baseline implemented in [1] optimised the parameters for a TIES-Merging [29] with DARE [31] between the three seed models. CMA-ES used a population of 9 and it made 9,000 evaluations, while here we run NN and GA with an archive size of 20 but limited them to 4,000 evaluations.

**Evaluation.** A correct answer must meet the following criteria: 1) the final numerical value must be correct, and 2) the reasoning text must be written in Japanese. To determine the language of the output, the library `fasttext` was used [17, 13].

### 4.2.2 Setup: Combining Math and Agentic Skills

**Models.** We combine a math specialist, `WizardMath-7B-V1.0` [16], with a specialist on agentic enviroments, `AgentEvol-7B` [28], to achieve an agent that performs well on the math benchmark GSM8k [5] and on the web shopping benchmark WebShop [30].

**Datasets.** For the math task, we used the test split of GSM8k as our test split (1319 samples). For the training split, we used the first 1319 samples of GSM8k train dataset. In the web shopping task, we used the WebShop environment implemented in [28]. The test split consistent of the first 100 tasks, while the training split were the next 100 tasks. We allowed the agents to take up to 7 steps.

**Baselines.** The CMA-ES optimised 32 mixing coefficients (one for each layer) for a SLERP merge between the two seed models. All methods were run for a 1000 evaluations on the training set. For the MAP-Elites we used two dimension to create a 4 by 4 grid: the accuracy on the math and on the web shopping training splits.

**Evaluation.** In this experiment, all methods used 1000 evaluations on the training set. NN and GA used an archive size of 15. CMA-ES used a population size of 25.

### 4.2.3 Results.

Tables 1 and 2 show that NN achieves the highest score. Both the matchmaker and the split-point techniques play a crucial role, however, the split-point seems to be slightly more important. Note that on Table 1 CMA-ES was run 2.25x longer than the other methods and used a more advanced merging technique (DARE-TIES), while on Table 2 all algorithms were run for the same amount of time and used the same merging method (SLERP). When combining the Math and Agentic skills, CMA-ES yielded a low score, likely due to suboptimal parameter partitioning, highlighting the need to include the merging boundaries in the optimization process.

Table 1: Accuracy of various methods on the Japanese Math benchmark MGSM-JA.

| Methods | MGSM-JA (acc ↑) |
|---|---|
| Shisa Gamma 7B v1 | 9.6 |
| WizardMath 7B v1.1 | 18.4 |
| Abel 7B 002 | 30.0 |
| **Natural Niches (NN)** | **54.4** |
| NN w/o matchmaker | 47.2 |
| NN w/o split-point | 44.4 |
| CMA-ES (DARE-TIES) | 52.0 |
| GA | 44.8 |
| LLama 2 70B | 18.0 |
| Japanese StableLM | 17.2 |
| GPT-3.5 | 50.4 |
| GPT-4 | 78.8 |

Table 2: Scores of various methods on math (GSM8k) and web shopping (WebShop) benchmarks.

| Methods | GSM8k (acc ↑) | WebShop (score ↑) | Average (score ↑) |
|---|---|---|---|
| WizardMath 7B v1.0 | 74.22 | 0.00 | 37.11 |
| AgentEvol 7B | 6.29 | 88.88 | 47.59 |
| **Natural Niches (NN)** | 40.74 | 86.17 | **63.46** |
| NN w/o matchmaker | 39.53 | 83.99 | 61.76 |
| NN w/o splitpoint | 33.31 | 87.91 | 60.61 |
| GA | 36.81 | 88.23 | 62.02 |
| MAP-Elites | 37.33 | 84.23 | 60.78 |
| CMA-ES (SLERP) | 46.21 | 43.49 | 44.85 |

### 4.2.4 Analysis

As shown in Figure 5, the findings from the MNIST dataset generalize to LLM merging. The Natural Niches method maintains high training coverage, as seen on the left side of the figure. The entropy rises early on as the models explore diverse niches (right), followed by a gradual decrease as low-performing models are removed, and the strengths of the models are aggregated. In contrast, MAP-Elites focuses on maximizing entropy at the cost of training efficiency and coverage, as it retains low-performing models. GA quickly reduces both coverage and entropy as it greedily converges on its top solution, ultimately collapsing the entire archive onto a single solution, with entropy nearing zero.

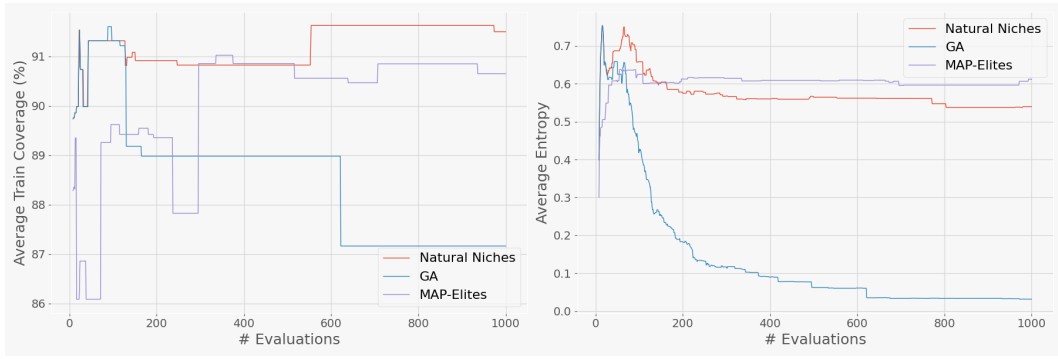

Figure 5: Left, the percentage of training data points that can be correctly labeled by at least one model in the population, averaged on the Math and Web shopping datasets. The right plot shows how the diversity in the performance of the population evolves with training.

## 5 Limitations & Future Work

The feasibility of model merging strongly depends on the degree of similarity between models. As demonstrated in [31], when fine-tuned models deviate significantly from their base models—often due to extensive, divergent training—merging becomes impractical. We hypothesize that models with divergent *state representations* are incompatible for merging. However, a standardized metric for model *compatibility* has yet to be established. Defining such a metric could allow it to be used as a form of regularization during preprocessing (e.g., fine-tuning), enabling better control over model compatibility and ensuring the success of merging.

Moreover, we believe there is a strong evolutionary pressure for models that are co-evolving together to remain compatible for merging. Should one model, diverge and become incompatible with others, it would no longer produce viable offspring, halting its improvement and leading to its eventual extinction. Testing this hypothesis through further research would provide valuable insights into the dynamics of model co-evolution.

Finally, incorporating a *compatibility* metric into the matchmaker heuristic could facilitate the co-evolution of distinct *species* of models, defined as groups that can merge with one another but not with others.

## 6 Conclusion

We've shown that model merging can significantly speed up the evolution of image classifiers when combined with a diversity-preservation technique. Moreover, this technique scales up to pre-trained LLMs. Our ablation studies show that both our proposed matchmaker heuristic and the use of crossover with split-point significantly improve the performance of the proposed method and have the potential to improve other evolutionary algorithms that use the crossover operation.

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
