# OpenReview forum: "Improving Model Merging with Natural Niches"
_NeurIPS.cc/2024/Workshop/UniReps — UniReps_

### Official Review · Reviewer_oLKR · 2024-09-30
**Natural Niches: evolutionary algorithm for model merging**

**Rating:** 4
**Confidence:** 3

**Review:**

This paper proposes a method for model merging using an evolutionary algorithm. In particular, authors claim three features: dynamic adjustment of merging boundaries,  a diversity preservation mechanism, and a heuristic-based mate selection strategy.
Weaknesses:
1. (2) assumes that model A and B should have compatible parameters for concatenation. Also, merging w/o boundaries can be  computationally-demanding
2. (3) is just the weighted and normalized version of (1) which is an incremental contribution
3. Downsampled MNIST experiments show extremely slow evolution process (15 hrs)
4. Extensive usage of vague notions such as "diversity" instead of scientific ones such as entropy
5. Code for language experiments is not available

---

### Official Review · Reviewer_Mhn8 · 2024-10-06
**This paper designs a merging method using an evolutionary algorithm. Benefiting from the dynamic adjustment of merging boundaries and some evolutionary techniques, the experiments achieve SOTA performance.**

**Rating:** 7
**Confidence:** 4

**Review:**

##### Summary of Strengths

- The writing is clear and easy to follow.
- The use of a evolutionary algorithm for model merging is both interesting and novel.
- The effectiveness of the method is validated through experiments on Evolving MNIST classifiers and LLM merging.

##### Summary of Weaknesses

- As noted by the authors, the mutation operator is not particularly effective in LLM merging.
- While the paper includes ablation studies, it offers limited comparisons with other methods, only comparing with CMA-ES.
- Although faster than CMA-ES, the computational cost seems high compared to other merging methods.
- There is a minor citation error: "Ties-merging" should be cited as NeurIPS 2023.

---

### Official Review · Reviewer_BJgp · 2024-10-06
**Improving Model Merging with Natural Niches**

**Rating:** 9
**Confidence:** 4

**Review:**

# Paper Summary
The paper addresses the model merging problem: combining multiple (possibly pre-trained) neural network parameters into an optimal one that inherits capabilities from its predecessors. The authors introduce _Natural Niches_ (NN), an evolutionary algorithm that dynamically adjusts merging boundaries, preserves diversity among models, and uses a heuristic for mate selection to optimize model merging. The paper builds upon previous work on model merging and genetic algorithms; particularly, the work by [Akiba _et al_, 2024](https://arxiv.org/abs/2403.13187) required manual partitioning of model parameters and static merging boundaries. The authors extend this by automating the exploration of merging boundaries and preserving diversity more effectively using techniques inspired by evolutionary processes.

Regarding the presented method, the algorithm iteratively selects two parent models using a matchmaker heuristic, which prioritizes complementarity in their strengths and then performs a parameter-space crossover using spherical linear interpolation (SLERP) at a randomly chosen split-point. This merging process produces new offspring models that combine the characteristics of both parents. This merging process produces new offspring models that combine the characteristics of both parents. The algorithm preserves population diversity by introducing competition for resources during fitness evaluation, ensuring diverse solutions are maintained. It replaces the worst-performing models in the archive with new offspring if they outperform existing ones, and over multiple generations, the algorithm progressively explores more complex merging boundaries, leading to the discovery of high-performing, merged models.

## Contributions
- A novel evolutionary algorithm called Natural Niches that includes:
    - a diversity preservation mechanism that enhances the effectiveness of merging models, and
    - a heuristic mate selection strategy to optimize the combination of models.
- Experiments include:
    - evolving small MNIST classifiers from scratch;
    - evolving pre-trained large language models (LLMs), showing state-of-the-art performance;
    - comparison with baselines, such as CMA-ES and MAP-Elites

## Quality, Originality, Significance
The authors' line of work is very relevant within the model merging community and has interesting implications when it comes to understanding how different parameter spaces relate to each other. The experiments are exhaustive, that is, they try to address all of the claims posed by the authors in the introduction. While the proposed research question seems to be a follow-up from a very recent paper, the authors do a good job stating how the question they address contributes to the state-of-the-art.

## Clarity
The paper is well written, following the NeurIPS format and sticking to the workshop's restrictions. Relevant related work is presented about model merging and evolutionary algorithms. However, the method section seems a bit short, for the number of details that the reader needs to understand to fully grasp the proposed algorithm.

# Pros
- The results have similar conclusions when it comes to different scales of model archives; this gives a pathway to extend the results to multi-model merging.
- Almost all experiments come with ablations that help to assess how both matchmaking and splitting weights contribute to NN.
- The authors provide a path to follow-up this work and hint its applicability to other related areas of machine learning.

# Cons
- The method’s feasibility seems to heavily depend on the similarity of models being merged.
- Big open questions are left on how to thoroughly address the time and space complexity of the algorithm.
- Experiments leave open the question on how would these evolutionary approaches compare to fine-tuning merged models.

# Further Questions and Suggestions
- The methodology section presents the main parts of the NN algorithm as 3 subsections; yet it seems that it is up to the reader to put everything together to fully describe the algorithm from start to end. Hence, I would suggest to add another descriptive figure of the pipeline, a pseudo-code block, or even a small section that puts the contents of 3.1, 3.2, and 3.3 together.
    - For instance, it is not clear how does the algorithm go from 1 split-point to many boundaries (Fig. 1) by looking at Eqs. 2, 3, and 4.
- While SLERP interpolations seem to preserve certain geometric properties, a small question is left on how their usage compares to (Euclidean) linear interpolations -- which often appear within model merging literature. Therefore, adding an ablation would be nice to address this concern; although I would leave this to the Appendix section, for the purposes of this workshop. Additionally, it could help pointing out the preference of SLERPs withing the evolutionary merging literature, as well.
- Do you have an explanation/intuition on why would evolving models from scratch achieve better performance than starting from pre-trained models (Fig. 2)? I am assuming pre-trained models are optimal so I am curious to know how this result relies on the complexity of the task itself (MNIST).
- Could you clarify the statement on not using mutations on pre-trained models because _"random alterations don't scale well to larger models"_?
- Could you clarify if the _training coverage_ concept relates to ensembling accuracy (from all the archive)? In any case, do we expect it to be a hard upper-bound even for the best performing NN-generated archive?

All in all, it is a very interesting approach to model merging, with a nice set of experiments. I look forward to discussing more with the authors at the workshop :).

---

### Official Review · Reviewer_zvDM · 2024-10-07
**Method for model merging with impressive results on LLMs**

**Rating:** 7
**Confidence:** 3

**Review:**

### Strengths:

1.
The paper is well-written, easy to read and proposes a nature-inspired model merging performance with solid results on  LLMs.

2.
The idea of splitting the layer for model merging is intuitive and novel.

3. Experiments in section 4.1.3 clearly shows that the proposed method (NN) helps maintain a more diverse set of model archive, showing the efficacy of the method.

4. The method is scalable, and strong results on different language tasks using LLMs make a strong case in favour of the proposed method.


### Weaknesses and suggestions:
1. Sections 3.1 and 3.2 are currently difficult to understand, and notations can be further improved to make it easier for readers.


2. Since the method is inspired by the genetic algorithm, a more detailed background would make it easier to follow the intuition behind the method.

3. Eqn 3 is not clear --- over which variable the equation is maximized?

4. More detailed discussion is required to explain why pre-trained models perform worse in Fig 2.

---

### Decision · Program_Chairs · 2024-10-10

**Decision:**

Accept

**Comment:**

In light of the positive reviewers' feedback and relevancy of the submission, we are pleased to accept this paper for presentation at UniReps 2024. We kindly ask the authors to incorporate the reviewers' suggestions and feedback in the final camera-ready version of the manuscript.